# The Association between Acetabulum Fractures and Subsequent Coxarthrosis in a Cohort of 77 Patients—A Retrospective Analysis of Predictors for Secondary Hip Osteoarthritis

**DOI:** 10.3390/jcm12206553

**Published:** 2023-10-16

**Authors:** Rafał Wójcicki, Tomasz Pielak, Jakub Erdmann, Piotr Walus, Bartłomiej Małkowski, Jakub Ohla, Łukasz Łapaj, Michał Wiciński, Jan Zabrzyński

**Affiliations:** 1Department of Orthopaedics and Traumatology, Faculty of Medicine, J. Kochanowski University in Kielce, 25-001 Kielce, Poland; ralfw@wp.pl (R.W.); tomasz.pielak@gmail.com (T.P.); walus.md@gmail.com (P.W.); zabrzynski@gmail.com (J.Z.); 2Department of Orthopaedics and Traumatology, Faculty of Medicine, Collegium Medicum in Bydgoszcz, Nicolaus Copernicus University in Toruń, 85-092 Bydgoszcz, Poland; jakub.ohla@wp.pl; 3Department of Urology, Oncology Centre Prof. Franciszek Łukaszczyk Memorial Hospital, Bydgoszcz, dr I. Romanowskiej St., 85-796 Bydgoszcz, Poland; malkowski.b@gmail.com; 4Department of General Orthopaedics, Musculoskeletal Oncology and Trauma Surgery, Poznan University of Medical Sciences, 61-701 Poznan, Poland; esperal@o2.pl; 5Department of Pharmacology and Therapeutics, Faculty of Medicine, Collegium Medicum in Bydgoszcz, Nicolaus Copernicus University, M. Curie 9, 85-090 Bydgoszcz, Poland; wicinski4@wp.pl

**Keywords:** acetabular fractures, hip osteoarthritis, post-traumatic arthritis, total hip arthroplasty, Kocher–Langenbeck approach

## Abstract

Objective: the aim of this study was to document the occurrence of THA after acetabulum surgery and examine the factors that predict its occurrence. Methods: This study included 77 consecutive patients who were admitted for acetabulum fracture surgery between 2012 and 2019. The inclusion criteria were acetabular fractures and indications for operative management. The exclusion criteria were acetabular fractures treated non-operatively, fractures requiring primary THA, and periprosthetic acetabular fractures. Data concerning demographics, date of injury, date of surgery, surgical approach, stabilization, and further reconstructive surgery were collected retrospectively. The number of patients who underwent THA and their risk factors were recorded. The minimum follow-up for each patient was 2 years of observation. A total of 77 patients with a mean age of 53 years were included. Results: At a mean follow-up of 2 years, THA was performed in 16 (20.8%) patients due to post-traumatic arthritis. An analysis of the surgical approaches showed that the Kocher–Langenbeck approach increased the risk of THA nearly 12 times compared with the ilioinguinal approach (*p* = 0.016). Furthermore, the duration of the waiting period for surgery significantly impacted the occurrence of THA, with each additional day leading to an 89% increase in the risk of prosthesis usage (*p* = 0.001). Conclusions: This study suggests that acetabular fractures may lead to post-traumatic hip osteoarthritis. The surgical approach and the waiting time for surgery are potential factors that may predict secondary hip osteoarthritis and the need for subsequent THA. However, further investigations should be performed to establish predictors for secondary hip osteoarthritis, and especially to determine the impact of the surgical approach.

## 1. Introduction

Acetabular fractures are among the most complex injuries in traumatology. They are most often the result of high-energy forces in the range of 2000–10,000 N, and they account for about 3% of all fractures [1,2]. The most common causes are traffic accidents and falls from height—76–89% and 7–20% of all such injuries, respectively [3]. In the population over 65 years of age, the frequency of acetabular fractures reaches approximately 12–14% [4]. The trauma’s energy acts on the area of the acetabulum and femoral head, and this can lead to post-traumatic changes that affect the distribution of forces and the biomechanics of the hip joint [2]. The consequences are accelerated degenerative changes in the joint tissues that require early hip arthroplasty, and these affect 19% of patients in the first two years after injury and 38% of patients after 5 years [5,6]. The mechanism of the relationship between acetabular cup fracture and subsequent arthritis is inconsistent. The development of post-traumatic hip osteoarthritis is caused by a range of factors, including cartilage and bone damage, changes in synovial fluid, increased chondrocyte metabolism, or an extended surgical approach [7,8,9,10]. Fractures of the acetabulum with displacements less than 2 mm have the potential to heal with good clinical outcomes. However, the newly formed tissue is structurally more like fibrocartilage than the physiological hyaline cartilage of the hip, and this is related to the increased instability of the femoral head [11].

In cases of stable fractures with displacement less than 2 mm, conservative management is the first choice, and rest, physiotherapy, analgesic treatment, and subsequent lifestyle modifications are recommended [2]. On the other hand, surgical treatment is necessary for treating unstable fractures and typically involves either open reduction and internal fixation (ORIF) or total hip arthroplasty (THA) [2,12]. Although numerous surgical approaches have been developed, the Kocher–Langenbeck (K-L) and ilioinguinal approaches continue to be the primary choices [13]. The K-L approach enables direct access to the outer surface of the posterior wall and the posterior column, as well as indirect access to the superior wall and the quadrilateral surface. To achieve this, the muscle fibers of the gluteus maximus are split and the short external rotators are detached [14]. The ilioinguinal approach is based on the detachment of the abdominal muscles and iliacus from the anterior two thirds of the iliac crest. It provides direct access to the internal iliac fossa, the anterior sacroiliac joint, the upper part of the pelvic brim, the quadrilateral surface, the superior pubic ramus, and the symphysis [15]. Most often, the K-L approach is the first choice for posterior column fractures, whereas the ilioinguinal approach is the first choice for anterior column fractures [16]. However, both can be chosen simultaneously in complex acetabular fractures [16,17]. 

The aim of this study was to investigate (1) the association between acetabulum fractures and subsequent coxarthrosis and (2) potential predictors for secondary hip osteoarthritis in a consecutive 77-patient cohort.

## 2. Materials and Methods

### 2.1. General Characteristics

This study comprised 77 consecutive patients admitted for isolated acetabulum fracture surgery between 2012 and 2019. The data were collected retrospectively in a single trauma center. 

The inclusion criteria were isolated acetabulum fractures and indications for operative treatment with primary osteosynthesis (displacement of articular surface > 2 mm, unstable fracture pattern, e.g., posterior wall fracture involving >40–50%, marginal impaction, intra-articular loose bodies, irreducible fracture dislocation). We excluded patients who had acetabular fractures that were treated non-operatively, fractures that required primary THA, periprosthetic acetabular fractures, and concomitant pelvic ring fractures. 

### 2.2. Fracture Classification

The acetabulum fractures were classified according to the Letournel and Judet system (A + T—anterior column with posterior hemi-transverse fracture; AC—anterior column; BC—both columns; PC—posterior column; PC + W—posterior column + posterior wall; PW—posterior wall; T—transverse; T + P—transverse with posterior wall fracture), while the pelvic ring fractures were classified according to the Young and Burgess system (LC—lateral compression; APC—anterior–posterior compression; VS—vertical shear).

### 2.3. Imaging

On admission, the patients underwent pelvic X-rays with additional Judet’s views. Moreover, each patient had a computed tomography (CT) evaluation on admission as well. 

### 2.4. Surgery

All patients were treated operatively using De Puy Synthes implants for pelvic fixation (a dedicated system for reconstructive pelvic and acetabular surgery) and 3.5 mm reconstruction plates with low profiles. The anesthesia was performed using general anesthesia due to the complicated nature of the cases and the often prolonged surgery. The ilioinguinal and Kocher–Langenbeck surgical approaches were used for the acetabulum fractures, depending on the extent and location of the specific fracture. The ilioinguinal approach was preferred in the anterior wall and anterior column fractures, both column fractures, and the posterior hemi-transverse fractures. The Kocher–Langenbeck approach was preferred in the posterior wall and posterior column fractures and in most of the transverse and T-shaped fractures. 

### 2.5. The Follow-Up

The minimum follow-up for each patient was 2 years of observation. The following demographic data were collected: age (years), sex, type of fracture, date of injury and date surgery, surgical approach and stabilization, further reconstructive surgery, and complications. During the follow up observation, two groups were distinguished: (1) patients who developed secondary hip osteoarthritis after fixation and required THA; and (2) patients without secondary osteoarthritis. The radiological outcomes were assessed at six months, one year, two years, and annually using pelvic X-rays. If osteoarthrosis was identified, a CT scan was performed in each case for operative planning. The scale used to describe the osteoarthritis of the hip observed in the X-rays was the Kellgren–Lawrence classification system. Each radiograph was assigned a grade from 0 to 4, and these correlated to increasing OA severity, with Grade 0 signifying no presence of OA and Grade 4 signifying severe OA. 

Indications for total hip arthroplasty in the studied group were as follows: clinical symptoms of degenerative joint disease (hip pain that limits everyday activities, such as walking or bending, hip pain that continues while resting, either day or night, and stiffness in a hip that limits the ability to move or lift the leg and limits the range of motion), inadequate pain relief from anti-inflammatory drugs, posttraumatic degenerative joint disease, or osteonecrosis with cartilage destruction seen in the radiographs. 

The total hip arthroplasties were carried out using the anterior mini-invasive approach (DAA) or the classical Kocher–Langenbeck approach. Moreover, if there was no need to remove the plates, the hip arthroplasty was carried out parallel without removing the osteosynthesis implants. 

### 2.6. Ethics

This study was performed in accordance with the Declaration of Helsinki for experiments involving humans after permission was received from the local Bioethics Committee (approval number: KB 645/2022). 

### 2.7. Statistical Analysis

Nominal variables were characterized according to the number of observations and their structures. A comparison between the two groups was conducted using Pearson’s chi-square test with or without Yates’s continuity correction, depending on the values expected for adequate contingency tables.

Numerical variables were described using basic statistics, i.e., mean and standard deviation in the case of a normal distribution and median plus range for an abnormal distribution. Their distributions were verified using Shapiro–Wilk tests, skewness coefficients, and kurtosis, as well as visual assessments of histograms. Comparisons of the variables between the groups were conducted using Welch *t*-tests for the independent groups and Mann–Whitney U tests, depending on the normality of their distributions. The homogeneity of variance was verified using Levene’s test. Logistic regression was performed to assess which parameters had an influence on the risk of prothesis after the hip fracture surgery and how the significant parameters affected the risk level. The following parameters were assessed: sex, age, type of fracture, surgical approach, and surgery waiting time. A two-step approach was used: firstly, univariate models were used for each variable in order to filter the significant variables. Secondly, multivariable logistic regression was run (only for parameters with *p* < 0.25 in the first step) to reveal the final set of significant parameters in a joint environment and understand their final impact. The quality of the multivariate model was determined using a chi-square test, a Hosmer and Lemeshow GOF test, and R^2^ Nagelkerke. All calculations were run assuming an *alpha* = 0.05 significance level, and the tests were two-sided. Analyses were conducted using R statistical package (version 4.1.2).

## 3. Results

The analyzed group consisted of 77 patients who had undergone hip fracture surgery between 2012 and 2019. Out of the total group, 21 (27.3%) were female and 56 (72.7%) were male (Table 1). The treatment of 16 (20.8%) patients out of the group was finalized with a prothesis within 2 years’ observation. The remaining patients did not develop hip degeneration and required no prothesis (79.2%). The mean time from initial trauma pelvic surgery to THA was 2 years and 3 months. Among the patients with a prothesis, there was sex parity, and in the group with no prothesis, males were the majority (78.7%). The sex difference between the groups was statistically significant: for male (vs. female), *p* = 0.048. The average patient age was 53.38 ± 18.07 years. The patients who underwent THA were older than the patients without who did not undergo THA: 65.30 years (±11.58) vs. 50.26 years (±18.23), respectively (*p* < 0.001) (Figure 1). 

Nearly half of the group were diagnosed with fractures of both columns (anterior + posterior) of the acetabulum (44.2%), and slightly less had transverse fractures (37.7%). The posterior type of fracture was the least common (18.2%). The fracture type did not differ significantly in the THA and non-THA groups (*p* = 0.827). Moreover, the K-L approach was used in slightly more than half of the cases (51.9%) and the ilioinguinal approach was used in 48.1% of the cases. However, the surgical approach distinguished the THA and non-THA groups in a significant way: K-L method (vs. Ilioinguinal), *p* = 0.038. The median waiting time for all patients between diagnosis and the surgical treatment of their fracture was 5.1 (range, 0–15.33) days. In the subgroups, the median waiting time was 8.08 and 4.00 days for THA and non-THA, respectively (*p* < 0.001).

Surgical approach and waiting time had a significant impact on the necessity of THA in both the univariate and multivariate models. An analysis of the surgical approach revealed that the K-L approach increased the risk of THA nearly 12 times (vs. Ilioinguinal approach): *OR* = 11.88 *CI*_95_ [1.98; 121.52], *p* = 0.016. Moreover, the waiting time for surgery had a significant influence on the risk of THA, and each additional day resulted in an 89% higher risk of prothesis usage: *OR* = 1.89 *CI*_95_ [1.37; 3.05], *p* = 0.001 (Figure 2). Sex and age had significant impacts only in the univariate models, with lower risk in males vs. females, *OR* = 0.27 *CI*_95_ [0.08; 0.86], *p* = 0.027, and increased risk with age, *OR* = 1.06 *CI*_95_ [1.02; 1.11], *p* = 0.006. The type of fracture did not significantly affect the risk of THA in either of the regression steps (Table 2).

The multivariate model evaluation included a chi-square test to verify the joint significance of all the variables (*p* < 0.001) and a Hosmer and Lemeshow GOF test to assess model fit (*p* = 0.964). Both tests confirmed the good quality of the model. Additionally, R2 Nagelkerky was calculated (62%), and its level proved to be relatively high, confirming the sufficient fir of the model.

Additionaly, X-rays and CT scans of particular patients presenting their fractures and results of performed surgeries were presented in Figure 3, Figure 4, Figure 5 and Figure 6.

## 4. Discussion

In this study, the association between surgically treated acetabular fractures and subsequent coxarthrosis was presented. Moreover, predictors for secondary hip osteoarthritis in the examined population were analyzed. In the studied patients, the incidence of secondary hip osteoarthritis was 20.8% within 2 years after the acetabular fracture and ORIF. This result is consistent with the reports of Letournel et al., Tannast et al., and Clarke-Jenssen et al., who presented 18%, 21%, and 18% (respectively) occurrences of secondary hip osteoarthritis [11,18,19]. Additional reports have indicated that the mean incidence rate is 20%, whereas in patients over the age of 65, this rises to approximately 28% [20,21]. In their comprehensive analysis, Tzu-Chun Chung et al. reported that the frequency of THA procedures was 7.28% for pelvic fractures, 17.82% for acetabular fractures, and 18.01% for combined acetabular and pelvic fractures [22]. On the other hand, lower occurrences was found by Henry et al. who analyzed the population of the Ontario region and reported a THA rate of 13.9% after a median of 6.25 years [12]. 

The relationship between chosen surgical approach and post-traumatic hip osteoarthritis has been widely discussed. The results of this study indicate that the use of the K-L approach during the ORIF of an acetabular fracture was a statistically significant risk factor for the development of secondary hip osteoarthritis. The posterior approach was chosen more often for acetabular fusion than the anterior approach (51.9 vs. 48.1%). The risk was nearly 12 times higher for the posterior approach compared with the anterior approach. The K-L approach is mainly used in fractures involving the posterior column. This pattern of fracture has been associated with a failure rate of approximately 26%, and it is also correlated with lower functional ability scores [23]. Moreover, Matta et al. concluded that comminuted posterior wall fractures have the highest rate of poor outcomes because of extended articular cartilage damage [24]. A similar conclusion was stated by Kreder et al., who found that even anatomical reduction alone did not prevent the development of arthritis and did not efficiently satisfy functional recovery in comminuted posterior wall fractures [25]. It is worth mentioning that Tannast et al. reported a 76% survival rate in patients with posterior acetabular wall fractures over a 20-year period [9], whereas Pantazopolous et al. reported an 85% survival rate after a mean follow-up of 15 years, and Chiu et al. reported an 81% survival rate after a mean follow-up of 7 years [26,27]. Additionally, Missonis et al. studied patients with posterior wall fractures and concomitant dislocations with a mean follow-up of 18.5 years. Positive clinical outcomes were found in 84.21% of the patients [28]. Although posterior wall fractures are associated with poor clinical prognoses, in this study, none of the acetabular fracture patterns were correlated with accelerated post-traumatic hip osteoarthritis. Thus, it can be concluded from this study that the K-L approach was a risk factor for post-traumatic hip osteoarthritis by itself. 

On the other hand, some surgeons claim that the ilioinguinal approach should be the favored approach. For instance, Mears et al. examined the use of the posterolateral, anterolateral, and extended lateral approaches in reconstructions and subsequent arthroplasties, depending on the type of fracture [29]. The authors emphasized the benefits of the use of the anterior approach, which enables stable fixation and the best fracture reduction. Similarly, Beaulé et al. recommended the anterior surgical approach, especially in cases of anterior column fracture [30]. However, according to data revealed by Mears et al., the link between the surgical approach and the development of post-traumatic hip osteoarthritis is ambiguous, and it is believed that surgeons should choose the approach with which they are most familiar. The experience of the surgical team is a crucial factor in minimizing failures [29]. Deficits in experience and skill can result in failure rates of up to 45% [31].

Patient age is another potentially significant factor in the development of post-traumatic hip osteoarthritis after an acetabulum fracture. However, the results of this study were not statistically significant, although the *p*-value was close to significance (*p* = 0.058 in the multivariate model). According to the available literature, the influence of age as a potential risk factor for post-traumatic hip osteoarthritis in patients with acetabular fractures is ambiguous. Some authors have noted a high rate of complications and the need for secondary arthroplasty in the first 2 years (up to 25%) after ORIF in elderly patients [32]. In another publication, which included a group of patients with a mean age of 67 years, the authors found that 30.95% of them required THA on average 5 years after ORIF [33]. Moreover, in their retrospective analysis of surgically treated acetabular fractures with a 2-year follow-up, Liebergall et al. indicated that older age was a statistically significant predictor of a negative outcome. Patients below 40 years of age had better prognoses, possibly due to the superior quality of their bone tissue and the fact that they had few comorbidities [34]. Likewise, Matta et al. conducted a study of 262 acetabular fracture patients with a 2-year follow-up. They found that 81% of the patients under 40 years old had good results, compared with 68% of patients aged 40 or older [24]. This study showed that the age variable is a parameter that has no significant impact in the multivariate model. However, the *p*-value was close to the limit of significance (*p* = 0.058) in our study. In addition, in the one-dimensional model, age turned out to be significant (*p* = 0.006). In summary, according to the abovementioned investigations, patient age remains questionable as a potential factor in the development of post-traumatic hip osteoarthritis. 

Another factor significantly influencing the development of degenerative changes is the waiting time until the final ORIF. In this study, each additional day of waiting increased the risk of developing hip osteoarthritis by 89%. In our studied population, surgery was performed as soon as possible after the injury. The influence of timing on acetabulum fracture surgery was also indicated by Letournel et al., who reported excellent results (80.69%) in a group of 492 patients treated surgically within 3 weeks of a fracture [2]. Similarly, Ziran indicated a higher rate of anatomical reduction in patients under 40 years of age treated less than 21 days after injury [35]. Furthermore, Meena et al. observed that 52% of patients treated after 2 weeks had poor clinical outcomes, while only 14% of patients who underwent surgery within two weeks experienced similarly unfavorable outcomes [36]. Achieving complete reduction can be challenging, especially in comminuted and old fractures of the acetabulum. The blood supply of the pelvis leads to abundant callus formation after 3 weeks post-injury, and this increases the difficulty of reduction [37].

In the available literature, information on the influence of gender on the occurrence of degenerative alterations in cartilage is limited. For instance, Meena et al. found that age had no impact on clinical outcome [36]. A comparison of the patients who were classified for THA (the same number of women and men) and those who were not (78.7% were men) revealed statistical differences close to the limit of significance. Gender had no significant effect on the THA after acetabular fracture in the multivariate model, but it was significant in the univariate model. 

In our study, we focused on an analysis of THA risk factors in a group of patients with acetabular fractures. In the group of patients with acetabular injuries, we performed only secondary THAs. As a rule, we did not remove the trauma implants prior to the fixation of the acetabular cup. However, in cases where they were an obstacle to acetabular implantation or there was a risk of concomitant infection, we decided to remove the implants as this was consistent with the guidance in the available literature [38]. Overall, we decided to remove the fixative material in four patients. Nevertheless, we never removed the implants in their entirety. However, there have been reports of the ORIF of acetabular fractures with simultaneous THA. This method was presented by Herscovici et al. [39]. It is most often recommended when there is no guarantee of stable and satisfactory positioning of the fragments, especially when the damage concerns the roof of the acetabulum and the ischium [40]. Therefore, a THA with the use of a custom-made implant offers a chance for effective reconstruction, limiting the next surgical intervention in the long term [41]. Therefore, it is recommended to use it simultaneously with an anastomosis, especially in the elderly [25,42]. The decision to use an anastomosis with simultaneous THA is difficult, as is the decision concerning the surgical approach. 

There are some limitations to this study. Firstly, this study was prone to various forms of bias due to its retrospective nature. Secondly, the size of the studied population was moderate, and this may have had an influence on the results. For instance, the sample size was not large enough to record a sufficient number of dislocations for statistical analysis. However, studies dealing with acetabulum problems are rare and based on various populations. Thirdly, there is still the potential for confounding due to unobserved factors and other comorbidities. Additionally, all the surgical procedures were performed by a single operative team. Despite the presented preferences in the choice of surgical approach, the decision still remains a matter of individual choice based on experience. A greater number of operators could have provided less biased results and offered a broader perspective on the issue of pelvic fractures. It is worth mentioning that an extended follow-up period could enhance the evaluation of the predictors. What is more, this study had no healthy control group with which to compare the patients.

## 5. Conclusions

Acetabulum fractures are devastating injuries for patients due to their association with secondary post-traumatic osteoarthritis. The incidence of secondary osteoarthritis of the hip after acetabular fracture was high (20.8%). The surgical approach and the waiting time for surgery were significant factors that could predict secondary hip osteoarthritis and the need for subsequent total hip arthroplasty. However, additional studies are necessary to provide a deeper understanding of post-traumatic hip osteoarthritis and establish predictors, particularly the surgical approach, for subsequent hip osteoarthritis in patients who suffer from acetabular fractures.

## Figures and Tables

**Figure 1 jcm-12-06553-f001:**
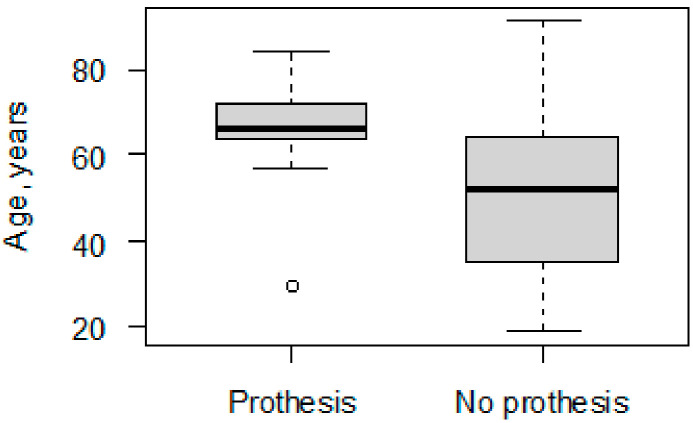
Boxplot of age distribution in groups with prothesis and without prothesis (*p* < 0.001).

**Figure 2 jcm-12-06553-f002:**
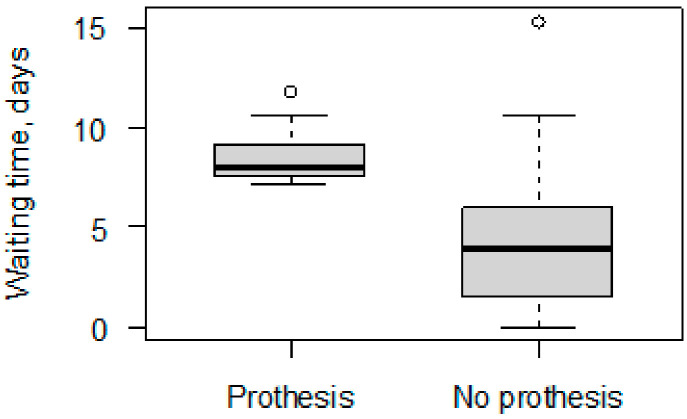
Boxplot of waiting time distributionin groups with prothesis and without prothesis (*p* < 0.001).

**Figure 3 jcm-12-06553-f003:**
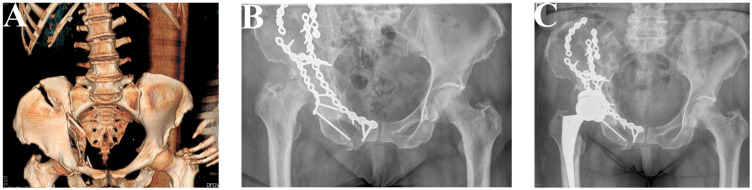
Patient 1. A 68-year-old woman suffered a comminuted acetabular fracture (**A**). Using the K-L and ilioinguinal approaches simultaneously, the fracture was reduced anatomically via a reconstruction plate (De Puy) (**B**). After 5 months, THA (P Pinacle, T Trilock De Puy) was performed using minimally invasive surgery via the direct anterior approach (MIS-DAA) (**C**).

**Figure 4 jcm-12-06553-f004:**
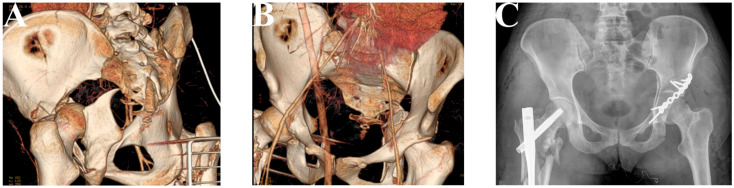
Patient 2. A 26-year-old woman had a transverse fracture and quadrilateral surface disruption of the left acetabulum (**A**,**B**). Employing the K-L approach, reduction was achieved using a reconstruction plate (De Puy) (**C**).

**Figure 5 jcm-12-06553-f005:**
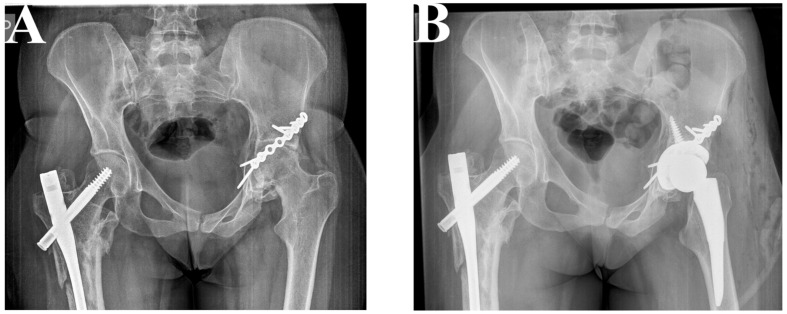
Patient 2. After 18 months of ORIF, X-rays showed post-traumatic hip osteoarthritis (**A**), and THA (P Pinacle, T Trilock De Puy) was performed using MIS-DAA (**B**).

**Figure 6 jcm-12-06553-f006:**
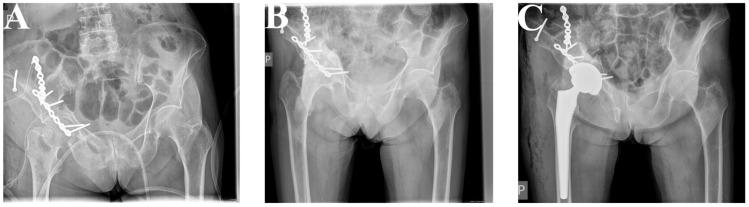
Patient 3. An 82-year-old woman suffered a transverse and posterior wall fracture of the right acetabulum (**A**). Using the K-L approach, reduction was achieved via a reconstruction plate (De Puy) (**B**). After 13 months of ORIF, X-rays showed post-traumatic hip osteoarthritis, and THA (P Pinacle, T Corail De Puy) was performed using MIS-DAA (**C**).

**Table 1 jcm-12-06553-t001:** Characteristics of studied group and comparisons between subgroups.

Variable	Total	THA	Non-THA	*MD* ^5^/*RR* ^6^ (95% *CI*)	*p* *
*N*	77	16	61		
Sex, *n* (%)					
Female	21 (27.3)	8 (50)	13 (21.3)	*reference*	0.048 ^1^
Male	56 (72.7)	8 (50)	48 (78.7)	0.38 (0.16; 0.87) ^6^	
Age, years, mean ± *SD*	53.38 ± 18.07	65.30 ± 11.58	50.26 ± 18.23	15.05 (7.51; 22.58) ^5^	<0.001 ^3^
Type of fracture, *n* (%)					
Both columns	34 (44.2)	8 (50.0)	26 (42.6)	*reference*	0.827 ^1^
Transverse	29 (37.7)	5 (31.2)	24 (39.3)	0.73 (0.27; 1.99) ^6^	
Posterior	14 (18.2)	3 (18.8)	11 (18.0)	0.91 (0.28; 2.94) ^6^	
Surgical approach, *n* (%)					
Ilioinguinal	37 (48.1)	4 (25.0)	33 (54.1)	*reference*	0.038 ^2^
K-L	40 (51.9)	12 (75.0)	28 (45.9)	2.78 (0.98; 7.85) ^6^	
Time to surgery, days, median (range)	5.10 (0.00–15.33)	8.08 (7.21–11.81)	4.00 (0.00–15.33)	4.08 (2.71; 5.42) ^5^	< 0.001 ^4^

SD—standard deviation. Groups compared using Pearson’s chi-square test with Yates’s continuity correction ^1^ or without Yates’s continuity correction ^2^, Welch *t*-test for independent groups ^3^, Mann–Whitney U test ^4^. MD ^5^—mean or median difference (THA vs. non-THA group) with 95% confidence interval, RR ^6^—relative risk with 95% confidence interval. *reference*—reference category for risk ratio. * Comparison between THA and non-THA groups.

**Table 2 jcm-12-06553-t002:** Logistic regression outcomes for THA and non-THA group dependent variables. *OR*—odds ratio with 95% confidence interval.

Variable	Univariate Models			Multivariate Model		
	*OR*	95% *CI* for *OR*	*p*	*OR*	95% *CI* for *OR*	*p*
Sex, male	0.27	0.08 to 0.86	0.027	0.30	0.04 to 1.69	0.180
Age, years	1.06	1.02 to 1.11	0.006	1.05	1.00 to 1.11	0.058
Type of fracture, (transverse vs. both columns)	0.68	0.18 to 2.32	0.540	na	na	na
Type of fracture, (posterior vs. both columns)	0.89	0.17 to 3.75	0.875	na	na	na
Surgical approach, (K-L vs. ilioinguinal)	3.54	1.09 to 13.78	0.046	11.88	1.98 to 121.52	0.016
Waiting time, days	1.86	1.40 to 2.76	<0.001	1.89	1.37 to 3.05	0.001

## Data Availability

Data are unavailable due to privacy and ethical restrictions.

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
