# Peer review of "The Association between Acetabulum Fractures and Subsequent Coxarthrosis in a Cohort of 77 Patients—A Retrospective Analysis of Predictors for Secondary Hip Osteoarthritis"

_jcm, 2023, doi:10.3390/jcm12206553_

Round 1

Reviewer 1 Report

Dear Authors,

Thank you for the effort and effort you put into your research. I have read your research in detail. Although the research is important in terms of the subject, there are major deficiencies in my writing. I will re-evaluate your research after major deficiencies are resolved.

Abstract

Divide this section into sections under the subheadings background and objectives, method, results and conclusion and expand the summary. Especially the results and conclusion sections are very shallow.

Introduction

  Even though you start this section with general definitions, you should add a new paragraph, especially after the second paragraph, to reveal what the studies reveal and the unique value of your work.

After the purpose statement, please highlight the main hypotheses of your research.

method

Unfortunately the Method section is not acceptable. Detail this section by dividing it into subheadings such as patients and procedures. Organize surgical procedures, radiological imaging and all evaluations under separate headings. The Method section should be completely revised.

Results

Although the photographs and figures seem sufficient, the appearance of the paintings is bad. Completely revise the tables as appropriate and regularly.

Discussion

I think this part is enough.

Best

Minor editing of English language required

Author Response

Dear Authors,

 Thank you for the effort and effort you put into your research. I have read your research in detail. Although the research is important in terms of the subject, there are major deficiencies in my writing. I will re-evaluate your research after major deficiencies are resolved.

Dear Sir / Madam,

Firstly I would like to thank you for your feedback and relevant suggestions. I am sending you the manuscript with the adjustments made.

Abstract
Divide this section into sections under the subheadings background and objectives, method, results and conclusion and expand the summary. Especially the results and conclusion sections are very shallow.

We have divided the abstract section into subsections and revised the results and conclusions.

Introduction
Even though you start this section with general definitions, you should add a new paragraph, especially after the second paragraph, to reveal what the studies reveal and the unique value of your work.

 After the purpose statement, please highlight the main hypotheses of your research.

The main hypothesis has been highlighted. The extra pharagraph has been added

Method
Unfortunately the Method section is not acceptable. Detail this section by dividing it into subheadings such as patients and procedures. Organize surgical procedures, radiological imaging and all evaluations under separate headings. The Method section should be completely revised.

It has been revised.

Results
Although the photographs and figures seem sufficient, the appearance of the paintings is bad. Completely revise the tables as appropriate and regularly.

Perhaps, Microsoft Word converted them, resulting in a loss of quality. The pictures were sent as supplementary files in better quality.

The tables have been revised

Discussion
I think this part is enough.

Reviewer 2 Report

This study entitled “The association between acetabulum fractures and subsequent coxarthrosis in a 77 patients cohort – predictors for secondary hip osteoarthritis” seems to have been generally well executed and written. Furthermore, I believe that this paper will be of great interest to the readers. However, I have a few remarks that require authors attention. 

Title

Please add in title the type of the study, i.e., prospective or retrospective. 

Abstract

Try to write the better conclusion for your Abstract, i.e., the better take-home message for the readers.

Keywords

Please add some additional MESH keywords to readers more easily identify your research.

Introduction

Please state the clear hypothesis of your study at the end of Introduction.

2. Materials and Methods

2.1. General characteristics

Please describe briefly the anesthesia management during the surgical procedure.

Here you stated that “The data was collected prospectively in a single trauma center”, but in the study limitations you wrote that “the study was prone to various forms of bias due to its retrospective nature”. Please correct this to avoid the confusion among the readers.

2.2. Ethics

Please state the date when the Ethical approval was gained.

Did you register your study (e.g., ClinicalTrials.gov). If yes, please state the number of registration and the date of registration.

2.3. Statistical analysis

Please state what type of multivariable logistic regression you have performed (e.g., Forward, Backward, Enter method,…)

Why the sample size calculation was not done? 

3. Results

In the first sentence of the Results please add the time period of conducting the study.

Please make the subsections to improve the readability of the Results section.

Discussion

In the first sentence of the Discussion state the main finding of your study.

Author Response

This study entitled “The association between acetabulum fractures and subsequent coxarthrosis in a 77 patients cohort – predictors for secondary hip osteoarthritis” seems to have been generally well executed and written. Furthermore, I believe that this paper will be of great interest to the readers. However, I have a few remarks that require authors attention. 

Dear Sir / Madam,

Firstly I would like to thank you for your feedback and relevant suggestions. I am sending you the manuscript with the adjustments made.

Title

Please add in title the type of the study, i.e., prospective or retrospective. 

It has been corrected.

Abstract

Try to write the better conclusion for your Abstract, i.e., the better take-home message for the readers.

We have divided the abstract section into subsections to be more transparent. In our opinion, conclusions are relatively simple and further simplification could lead to imprecision.

Keywords

Please add some additional MESH keywords to readers more easily identify your research.

It has been added.

Introduction

Please state the clear hypothesis of your study at the end of Introduction.

It has been corrected.

  1. Materials and Methods

2.1. General characteristics

Please describe briefly the anesthesia management during the surgical procedure.

Here you stated that “The data was collected prospectively in a single trauma center”, but in the study limitations you wrote that “the study was prone to various forms of bias due to its retrospective nature”. Please correct this to avoid the confusion among the readers.

The anesthesia management has been considered and done.

The mistake has been corrected.

2.2. Ethics

Please state the date when the Ethical approval was gained.

Did you register your study (e.g., ClinicalTrials.gov). If yes, please state the number of registration and the date of registration.

The approval number from the local Bioethics Committee was included in the Ethics section. To our knowledge, the date of approval is not required by MDPI journal.

2.3. Statistical analysis

Please state what type of multivariable logistic regression you have performed (e.g., Forward, Backward, Enter method,…)

Why the sample size calculation was not done? 

All the statistical analyses were done by a professional biostatistics company.

The model is created in such a way that the dependent variable is the prosthesis (its use or not). And the rest of the variables are predictors (i.e. how the predictors influence the use of the prosthesis in all 77 patients studied).

P – if p < 0.05, then the relationship between a given parameter and our modeled variable (here, a prosthesis or lack thereof) is significant. So we only look at variables with p < 0.05.

OR (odds ratio) - the interpretation is as follows:

OR = 1 – no influence of a given variable on the need to use a prosthesis

OR > 1 – by what percentage does the risk of needing to use a prosthesis increase if a given parameter increases by 1 (for quantitative variables), or if a given parameter appears vs. will not appear (for nominal variables, e.g. in the case of the type of surgical access, this will be a risk if K-L vs. Ilioinguinal is used). For example, OR = 1.89 for the waiting time for the procedure means an increase in the risk of needing to use a prosthesis by 89% if the waiting time increases by 1 day.

OR < 1 – similarly to the above, but a decrease in the risk of using a prosthesis (in this analysis, we do not have OR < 1 results that would be significant).

For the full picture, I added a confidence interval for each OR.

In the case of dummy (categorical) variables, we would interpret the same OR (OR = 1.89) as follows: patients who experienced the phenomenon described by this variable had an 89% higher risk of needing a prosthesis compared to patients in whom this phenomenon did not occur, with a 95% probability that this increase in risk ranges from 37% to 205%.

Interpretation of model quality assessment parameters:

  1. Chi-square – we want p < 0.05.
  2. Nagelkerki's R2 - can have a value from 0 to 1. The higher the value, the better the model fits, i.e. the better it covers all the factors explaining the need to use a prosthesis. So in our case, R2 = 62% tells us that the variables that ultimately turned out to be important in assessing the risk of using a prosthesis explain the need for its use in approximately 62% and there are other factors that influence the need to use a prosthesis, but they were not taken into account in the model. At the same time, R2 = 62% for the logistic model is quite a high value.
  3. Hosmer and Lemeshow – here we want p > 0.05.

  1. Results

In the first sentence of the Results please add the time period of conducting the study.

Please make the subsections to improve the readability of the Results section.

We have added the time of period and pharagraphs to make it more transparent

Discussion

In the first sentence of the Discussion state the main finding of your study.

The main finding was stated in the first sentence.

Reviewer 3 Report

The authors studied the incidences of acetabulum fractures associated with secondary coxarthrosis in patients with hip OA.

Comments

1.      Fig.1 and 2 should be deleted as they duplicate the data of Table 1.

2.      Lines 243-245: These sentences are not clear. The authors should indicate what kind of factor they describe on line 243 and indicate to which part of the study p-value is referred.

3.      Lines 259-260: This sentence should be rephrased and p-value should be indicated.

4.      Lines

5.      276-278: This sentence should be added to paragraph 3 on page 8.

6.      Conclusion should be rewritten as it is not related to the description of authors findings.

Author Response

The authors studied the incidences of acetabulum fractures associated with secondary coxarthrosis in patients with hip OA.

Dear Sir / Madam,

Firstly I would like to thank you for your feedback and relevant suggestions. I am sending you the manuscript with the adjustments made.

Comments

  1. 1 and 2 should be deleted as they duplicate the data of Table 1.

We decided to present some features in boxplots as it display differences among groups in a clear way.

  1. Lines 243-245: These sentences are not clear. The authors should indicate what kind of factor they describe on line 243 and indicate to which part of the study p-value is referred.

Age of patient is being discussed on lines 243-260. We have added referrence to mentioned p-value – it comes from the multivariate model.

  1. Lines 259-260: This sentence should be rephrased and p-value should be indicated.

We have added p-value and more transparent summary.

  1. Lines
  2. 276-278: This sentence should be added to paragraph 3 on page 8.

In our opinion, the callus formation may explain difficulties in achieving correct reduction and subsequent hip osteoarthritis. Thus, these sentences match to the paragraph about waiting time till the final ORIF and risk of developing hip osteoarthritis.

  1. Conclusion should be rewritten as it is not related to the description of authors findings.

It has been revised.

Round 2

Reviewer 1 Report

The manuscript is good.

Best

Author Response

Thank you.